# ANALYZING THE EFFECTS OF CLASSIFIER LIPSCHITZNESS ON EXPLAINERS

## ABSTRACT

Machine learning methods are getting increasingly better at making predictions, but at the same time they are also becoming more complicated and less transparent. As a result, explainers are often relied on to provide interpretability to these *black-box* prediction models. As crucial diagnostics tools, it is important that these explainers themselves are reliable. In this paper we focus on one particular aspect of reliability, namely that an explainer should give similar explanations for similar data inputs. We formalize this notion by introducing and defining *explainer astuteness*, analogous to astuteness of classifiers. Our formalism is inspired by the concept of *probabilistic Lipschitzness*, which captures the probability of local smoothness of a function. For a variety of explainers (e.g., SHAP, RISE, CXPlain), we provide lower bound guarantees on the astuteness of these explainers given the Lipschitzness of the prediction function. These theoretical results imply that locally smooth prediction functions lend themselves to locally robust explanations. We evaluate these results empirically on simulated as well as real datasets.

## 1 INTRODUCTION

Machine learning models have improved over time at prediction and classification, especially with the advances made in deep learning and availability of large amounts of data. These gains in predictive power have often been achieved using increasingly complex and *black-box* models. This has led to significant interest in, and a proliferation of, *explainers* that provide explanations for the predictions made by these black-box models. Given the crucial importance of these explainers it is imperative to understand what makes them reliable.

In this paper we focus on explainer robustness. A robust explainer is one where similar inputs results in similar explanations (Alvarez-Melis & Jaakkola, 2018). For example, consider two patients given the same diagnosis in a medical setting. These two patients share identical symptoms and are demographically very similar, therefore a diagnostician would expect that factors influencing the model decision should be similar as well. Prior work in explainer robustness suggests that this expectation does not always hold true (Alvarez-Melis & Jaakkola, 2018; Ghorbani et al., 2019); small changes to the input samples can result in large shifts in explanation. For this reason we investigate the theoretical underpinning of explainer robustness. Specifically, we focus on investigating the connection between explainer robustness and smoothness of the black-box function being explained.

We propose and formally define *explainer astuteness* – a property of explainers which captures the probability that a given method provides similar explanations to similar data points. This definition allows us to evaluate the robustness for a given explainer over the entire dataset and helps tie explainer robustness to probabilistic Lipschitzness of classifiers. We then provide a theoretical way to connect this explainer astuteness to the *probabilistic Lipschitzness* of the black-box function that is being explained. Since probabilistic Lipschitzness is a measure of the probability that a function is smooth in a local neighborhood, our results demonstrate how the smoothness of the black-box function itself impacts the astuteness of the explainer. This implies that *enforcing smoothness on black-box functions lends them to more robust explanations.*

**Related Work.** A wide variety of explainers have been proposed in the literature (Guidotti et al., 2018; Arrieta et al., 2020). Explainers can broadly be categorized as feature attribution or feature selection explainers. Feature attribution explainers provide continuous-valued importance scores to each of the input features, while feature selection explainers provide binary decisions on whether

a feature is important or not. Some popular feature attribution explainers can be viewed through the lens of Shapley values such as SHAP (Lundberg & Lee, 2017), LIME (Ribeiro et al., 2016) and LIFT (Shrikumar et al., 2016). Some models such as CXPlain (Schwab & Karlen, 2019), PredDiff (Zintgraf et al., 2017) and feature ablation explainers (Lei et al., 2018) calculate feature attributions by simulating individual feature removal, while other methods such as RISE (Petsiuk et al., 2018) calculate the mean effect of a feature's presence to attribute importance to it. In contrast, feature selection methods include individual selector approaches such as L2X (Chen et al., 2018) and INVASE (Yoon et al., 2018), and group-wise selection approaches such as gI (Masoomi et al., 2020). While seemingly diverse, these models have been shown to have striking underlying similarities, for example, Lundberg & Lee (2017) unify six different explainers under a single framework. Recently, Covert et al. (2020) went a step further and combined 25 existing methods under the overall class of *removal-based explainers*.

Similarly, there has been a recent increase in research focused on analyzing the behaviour of these explainers themselves in ways similar to how classification models have been analyzed. Recent work has focused on dissecting various properties of explainers. Yin et al. (2021) propose stability and sensitivity as measures of faithfulness of explainers to the decision-making process of the black-box model and empirically demonstrate the usefulness of these measures. Li et al. (2020) explore connections between local explainability and model generalization. Ghorbani et al. (2019) test the robustness of explainers through systemic and adversarial perturbations. Agarwal et al. (2022) define and discuss theoretical guarantees around faithfulness and stability in the context of Graph Neural Networks. Our definition of astuteness is related to what they call stability, but defined as a probability over all available instances in such a way that connection to probabilistic Lipschitzness of the classifier becomes clear. Alvarez-Melis & Jaakkola (2018) empirically show that robustness, in the sense that explainers should provide similar explanations for similar inputs, is a desirable property and how forcing this property yields better explanations. Recently, Agarwal et al. (2021) explore the robustness of LIME (Ribeiro et al., 2016) and SmoothGrad (Smilkov et al., 2017), and prove that for these two methods their robustness is related to the maximum value of the gradient of the predictor function. Our work is closely related to Alvarez-Melis & Jaakkola (2018) and Agarwal et al. (2021) on explainer robustness. However, instead of enforcing explainers to be robust themselves (Alvarez-Melis & Jaakkola, 2018), our theoretical results suggest that ensuring robustness of explanations also depends on the smoothness of the black-box function that is being explained. Our results are complementary to the results obtained by Agarwal et al. (2021) in that our theorems cover a wider variety of explainers as compared to only Continuous LIME and SmoothGrad (see contributions below). We further relate robustness to probabilistic Lipschitzness of black-box models, which is a quantity that can be empirically estimated.

Additionally, there has been recent work estimating upper-bounds of Lipschitz constant for neural networks (Virmaux & Scaman, 2018; Fazlyab et al., 2019; Gouk et al., 2021), and enforcing Lipschitz continuity during neural networks training, with an eye towards improving classifier robustness (Gouk et al., 2021; Aziznejad et al., 2020; Fawzi et al., 2017; Alemi et al., 2016). Fel et al. (2022) empirically demonstrated that 1-Lipschitz networks are better suited as predictors that are more explainable and trustworthy. Our work provides crucial additional motivation for that line of research; i.e., it provides theoretical reasons to improve Lipschitzness of neural networks from the perspective of enabling more robust explanations.

**Contributions:**

- We formalize and define *explainer astuteness* which captures the probability that a given explainer provides similar explanations to similar points. This formalism allows us to theoretically analyze robustness properties of explainers.
- We provide theoretical results that connect astuteness of explainers to the smoothness of the black-box function they are providing explanations on. **Our results suggest that smooth black-box functions result in explainers providing more astute explanations**. While this statement is intuitive, proving it is non-trivial and requires additional assumptions for different explainers (See Section 3.2).
- Specifically we prove this result for astuteness of three classes of explainers: (1) Shapley value based (e.g. SHAP), (2) explainers that simulate mean effect of features (e.g. RISE), and (3) explainers that simulate individual feature removal (e.g. CXPlain). Formally, our theorems establish a lower bound on explainer astuteness that depends on the Lipschitzness

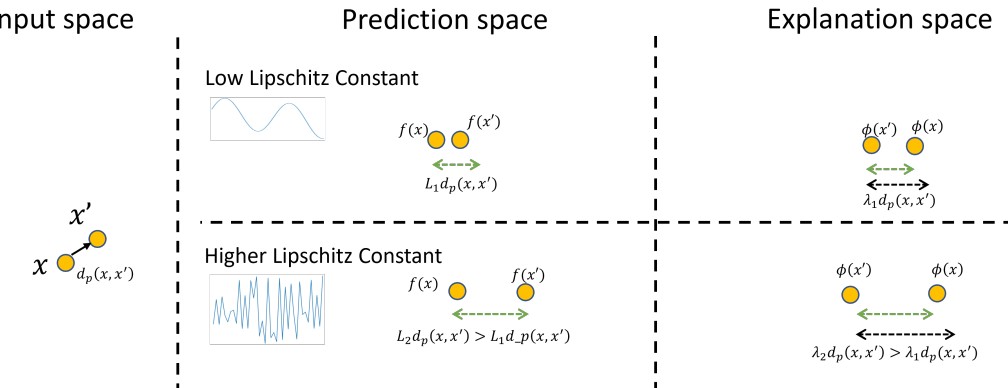

Figure 1: In this figure we visualize the implication of our theoretical results. For a black-box prediction function that is locally Lipschitz with a constant $L_1$, the predictions for any two points $x, x'$ such that $d_p(x, x') \leq r$ are within $L_1 d_p(x, x')$ distance from each other. Given such a prediction function, the explanation for the same data points are also expected to be within $\lambda_1 d_p(x, x')$ of each other where $\lambda_1 = CL_1\sqrt{d}$ where C is a constant. If we consider a second black-box function with $L_2 > L_1$ that results in $\lambda_2 > \lambda_1$, indicating that the explanations for this black-box function can actually end up being farther apart as compared to the first prediction function. This result implies that **locally smooth black-box functions lend themselves to more astute (i.e., robust) explanations.**

- of the black-box function and square root of data dimensionality. Figure 1 summarizes this main contribution of our work.
- We demonstrate experimentally that this lower bound indeed holds in practice by comparing the astuteness predicted by our theorems to the observed astuteness on simulated and real datasets. We also demonstrate experimentally that the same neural network when trained with Lipschitz constraints lends itself to more astute explanations compared to when it is trained with no constraints.

## 2 BACKGROUND AND NOTATIONS

### 2.1 REMOVAL-BASED FEATURE EXPLAINERS

As mentioned in Section 1, there exist a wide variety of explainers. Owing to this diversity, in this work, we concern ourselves with *removal-based feature attribution explainers* as defined by Covert et al. (2020) (which showed 25 existing methods under this umbrella). Removal based feature attribution explainers are methods that define a feature's influence through the impact of removing it from a model and assigning continuous-valued scores to each feature signifying its importance. This includes popular approaches such as SHAP and SHAP variants including KernelSHAP, LIME, DeepLIFT (Lundberg & Lee, 2017), mean effect based methods such as RISE (Petsiuk et al., 2018), and individual effects based methods such as CXPlain (Schwab & Karlen, 2019), PredDiff (Zintgraf et al., 2017), permutation tests (Strobl et al., 2008), and feature ablation explainers (Lei et al., 2018). All of these methods simulate feature removal either explicitly or implicitly. For example, SHAP explicitly considers effect of using subsets that include a feature as compared to the effect of removing that feature from the subset. RISE removes subsets of features while always keeping the feature that is being evaluated, and estimates the average effect of keeping that feature when other features are randomly removed. CXPlain explicitly considers the impact of removing a feature on the loss function used in training the predictor function.

### 2.2 NOTATION

We denote $d$-dimensional input data as $x \in \mathbb{R}^d$, from a data distribution $\mathcal{D}$. The black-box predictor function is denoted by $f$, where $f(x)$ is the prediction given $x$, this function is assumed to have been trained on the training samples from $\mathcal{D}$. The explainer is represented by a function $\phi$ where $\phi(x) \in \mathbb{R}^d$ is the feature attribution vector representing attributions for all features in $x$ while $\phi_i(x) \in \mathbb{R}$ is the attribution for the $i^{th}$ feature. To simulate the presence or absence of features in a given subset of features, we use an indicator vector $z \in \{0, 1\}^d$, where $z_i = 1$ when the $i^{th}$ feature is

present in the subset. To indicate we are only using subsets where feature $z_i = 1$, we use $z_{+i}$; and to indicate only using subsets where feature $z_i = 0$, we use $z_{-i}$. Lastly, the $p$-norm induced distance between any two points $x, x'$ is denoted by $d_p(x, x') = ||x - x'||_p$, where $||.||_p$ is the $p$-norm.

## 3    EXPLAINER ASTUTENESS

Our main interest is in defining a metric that can capture the difference in explanations provided by an explainer to points that are close to each other in the input space. The same question has been asked for classifiers. Bhattacharjee & Chaudhuri (2020) came up with the concept of *Astuteness* of classifiers, which captures the probability that similar points are assigned the same label by a classifier. Formally they provide the following definition:

**Definition 1.** *Astuteness of classifiers* (Bhattacharjee & Chaudhuri, 2020): The astuteness of a classifier $f$ over $\mathcal{D}$, denoted as $A_r(f, \mathcal{D})$ is the probability that $\forall x, x' \in \mathcal{D}$ such that $d(x, x') \leq r$ the classifier will predict the same label.

$$A_r(f, \mathcal{D}) = \mathbb{P}_{x,x' \sim \mathcal{D}}[f(x) = f(x')|d(x, x') \leq r] \tag{1}$$

The obvious difference in trying to adapt this definition of astuteness to explainers is that explanations for nearby points do not have to be *exactly* the same. Keeping this in mind, we propose and formalize *explainer astuteness*, as the probability that the explainer assigns *similar* explanations to similar points. The formal definition is as follows:

**Definition 2.** *Explainer astuteness*: The *explainer astuteness* of an explainer $E$ over $\mathcal{D}$, denoted as $A_{r,\lambda}(E, \mathcal{D})$ is the probability that $\forall x, x' \in \mathcal{D}$ such that $d_p(x, x') \leq r$ the explainer $E$ will provide explanations $\phi(x), \phi(x')$ that are at most $\lambda \cdot d_p(x, x')$ away from each other, where $\lambda \geq 0$

$$A_{r,\lambda}(E, \mathcal{D}) = \mathbb{P}_{x,x' \sim \mathcal{D}}[d_p(\phi(x), \phi(x')) \leq \lambda \cdot d_p(x, x') \,\big|\, d_p(x, x') \leq r] \tag{2}$$

A critical observation about definition 2 is that it not only relates to the previously defined notion of classifier astuteness, but also connects to the concept of *probabilistic Lipschitzness*. Probabilistic Lipschitzness captures the probability of a function being locally smooth given a radius $r$. It is specially useful for capturing a notion of smoothness of complicated neural network functions for which enforcing global and deterministic Lipschitzness is difficult. Mangal et al. (2020) formally defined probabilistic Lipschitzness as follows:

**Definition 3.** *Probabilistic Lipschitzness* (Mangal et al., 2020): Given $0 \leq \alpha \leq 1$, $r \geq 0$, a function $f : \mathbb{X} \to \mathbb{R}$ is probabilistically Lipschitz with a constant $L \geq 0$ if

$$\mathbb{P}_{x,x' \sim \mathcal{D}}[d_p(f(x), f(x')) \leq L \cdot d_p(x, x') \,\big|\, d_p(x, x') \leq r] \geq 1 - \alpha \tag{3}$$

### 3.1    THEORETICAL BOUNDS OF ASTUTENESS

A cursory comparison between equation 2 and equation 3 hints at the two concepts being related to each other. In fact, explainer astuteness can be viewed as probabilistic Lipschitzness of the explainer when it is viewed as a function with a Lipschitz constant $\lambda$. However, a much more interesting question to explore is how the astuteness of explainers is connected to the Lipschitzness of the black-box model they are trying to explain. We introduce and prove the following theorems which provide theoretical bounds that connect the Lipschitz constant $L$ of the black-box model to the astuteness of various explainers including SHAP (Lundberg & Lee, 2017), RISE (Petsiuk et al., 2018), and methods that simulate individual feature removal such as CXPlain (Schwab & Karlen, 2019).

#### 3.1.1    ASTUTENESS OF SHAP

SHAP (Lundberg & Lee, 2017) is one of the most popular feature attribution based explainers in use today. Lundberg & Lee (2017) unify 6 existing explanation approaches within the SHAP framework. Each of these explanation approaches (such as DeepLIFT and kernelSHAP) can be viewed as approximations of SHAP, since SHAP in its theoretical form is difficult to calculate. However, in this section we use the theoretical definition of SHAP to establish bounds on astuteness.

For a given data point $x \in \mathcal{X}$ and a prediction function $f$, the feature attribution provided by SHAP for the $i^{th}$ feature is given by:

$$\phi_i(x) = \sum_{z_{-i}} \frac{|z_{-i}|!(d - |z_{-i}| - 1)!}{d!}[f(x \odot z_{+i}) - f(x \odot z_{-i})] \tag{4}$$

Before moving on to the actual theorem, we introduce and prove the following Lemma which is necessary for the proof of Theorem 3.1.

**Lemma 1.** *If,*

$$\mathbb{P}_{x,x'\sim\mathcal{D}}[d_p(f(x),f(x')) \leq L*d_p(x,x') \,\big|\, d_p(x,x') \leq r] \geq 1-\alpha$$

*then for $y = x \odot z_{+i}, y' = x' \odot z_{+i}$, i.e. $y,y' \in \cup\mathbb{N}_k = \{y|y \in \mathbb{R}^d, ||y||_0 = k, y_i \neq 0\}$ for $k = 1, \ldots, d$*

$$\mathbb{P}_{x,x'\sim\mathcal{D}}[d_p(f(y),f(y')) \leq L*d_p(y,y') \,\big|\, d_p(y,y') \leq r] \geq 1-\beta$$

*where $\beta \geq \alpha$ assuming that the distribution $\mathcal{D}$ is defined for all $x$ and $y$ and the equality is approached if the probability of sampling points from the set $\mathbb{N}_k = \{y|y \in \mathbb{R}^d, ||y||_0 = k, y_i \neq 0\}$ approaches zero for $k = 2, \ldots, d$ relative to the probability of sampling points from $\mathbb{N}_1$.*

*Proof.* (Sketch, full proof in Appendix A)

Assume $p_k$ is the probability of occurrence of the set $\mathbb{N}_k = \{x|x \in \mathbb{R}^d, ||x||_0 = k, x_i \neq 0\}$ in the input space and $\gamma_k$ is the probability of the set of points that violate Lipschitzness in the set $\mathbb{N}_k$. In finite case each set $\mathbb{N}_k$ can be mapped to a set $\mathbb{N}'_k$ of cardinality $2^{d-k}|\mathbb{N}_k|$ after masking with all possible $z_{+i}$. In probability terms, the probability of $\mathbb{N}'_k$ can be written as $p'_k = \frac{2^{d-k}p_k}{\sum_{j=1}^d 2^{-j}p_j}$. Let $\beta$ be the the proportion of points in *all* $\mathbb{N}'_k$ that also violate Lipschitzness in their unmasked form then $\beta$ can be written as

$$\beta = \frac{\sum_{k=1}^d 2^{-k}p_k\gamma_k}{\sum_{j=1}^d 2^{-j}p_j}$$

Considering worse case $\beta$ requires solving the following equation,

$$\beta^* = \max_{\gamma_1,\ldots,\gamma_d} \frac{\sum_{k=1}^d 2^{-k}p_k\gamma_k}{\sum_{j=1}^d 2^{-j}p_j}, \sum_{i=1}^d p_i\gamma_i = \alpha, 0 \leq \alpha \leq 1, 0 \leq \gamma_i \leq 1, \forall i = 1, \ldots, d \quad (5)$$

The result of this maximization will be $\beta^* \geq \alpha$. In the specific case where $p_k \to 0$ for $k = 2, \ldots, d$ (i.e., where the probability of sampling any $x$ with a 0 valued element is 0), $\beta \to \alpha$. □

**Theorem 3.1.** *(Astuteness of SHAP) Consider a given $r \geq 0$ and $0 \leq \alpha \leq 1$, and a trained predictive function $f$ that is probabilistic Lipschitz with a constant $L$, radius $r$ measured using $d_p(.,.)$ and with probability at least $1-\alpha$. Then for SHAP explainers we have astuteness $A_{r,\lambda} \geq 1-\beta$ for $\lambda = 2\sqrt[p]{d}L$. Where $\beta \geq \alpha$, and $\beta \to \alpha$ under conditions specified in Lemma 1.*

*Proof.* Given input $x$ and another input $x'$ s.t. $d(x,x') \leq r$. And letting $\frac{|z_{-i}|!(d-|z_{-i}|-1)!}{d!} = C_z$. Using equation 4 we can write,

$$d_p(\phi_i(x),\phi_i(x')) = ||\sum_{z_{-i}} C_z[f(x \odot z_{+i}) - f(x \odot z_{-i})] - \sum_{z_{-i}} C_z[f(x' \odot z_{+i}) - f(x' \odot z_{-i})]||_p \quad (6)$$

Combining the two sums and re-arranging the R.H.S,

$$d_p(\phi_i(x),\phi_i(x')) = ||\sum_{z_{-i}} C_z[f(x \odot z_{+i}) - f(x' \odot z_{+i}) + f(x' \odot z_{-i}) - f(x \odot z_{-i})]||_p \quad (7)$$

Using triangular inequality on the R.H.S twice,

$$d_p(\phi_i(x),\phi_i(x')) \leq ||\sum_{z_{-i}} C_z[f(x \odot z_{+i}) - f(x' \odot z_{+i})]||_p + ||\sum_{z_{-i}} C_z[f(x' \odot z_{-i}) - f(x \odot z_{-i})]||_p$$

$$\leq \sum_{z_{-i}} C_z||f(x \odot z_{+i}) - f(x' \odot z_{+i})||_p + \sum_{z_{-i}} C_z||f(x' \odot z_{-i}) - f(x \odot z_{-i})||_p$$

$$(8)$$

We can replace each value inside the sums in equation 8 with the maximum value across either sums. Doing so would still preserve the inequality in equation 8, as the sum of $n$ values is always less than

the maximum among those summed $n$ times. Without loss of generality let us assume this maximum is $|f(x \odot z^*_{+i}) - f(x' \odot z^*_{+i})|$ for some particular $z^*$. This gives us:

$$d_p(\phi_i(x), \phi_i(x')) \leq ||f(x \odot z^*_{+i}) - f(x' \odot z^*_{+i})||_p \sum_{z_{-i}} C_z + ||f(x \odot z^*_{+i}) - f(x' \odot z^*_{+i})||_p \sum_{z_{-i}} C_z \quad (9)$$

However, $\sum_{z_{-i}} C_z = \sum_{z_{-i}} \frac{|z_{-i}|!(d-|z_{-i}|-1)!}{d!} = 1$, which gives us,

$$d_p(\phi_i(x), \phi_i(x')) \leq 2||f(x \odot z^*_{+i}) - f(x' \odot z^*_{+i})||_p = 2d_p(f(x \odot z^*_{+i}), f(x' \odot z^*_{+i})) \quad (10)$$

Using the fact that $f$ is probabilistic Lipschitz with a given constant $L \geq 0$, $d_p(x, x') \leq r$, $d_p(x \odot z^*_{+i}, x' \odot z^*_{+i}) \leq d_p(x, x')$ and Lemma 1. We get:

$$P[2d_p(f(x \odot z^*_{+i}), f(x' \odot z^*_{+i})) \leq 2L \cdot d_p(x, x')] \geq 1 - \beta$$

Since equation 10 establishes that $d_p(\phi_i(x), \phi_i(x')) \leq 2d_p(f(x \odot z^*_{+i}), f(x' \odot z^*_{+i}))$, the below inequality can be now established:

$$P[d_p(\phi_i(x), \phi_i(x')) \leq 2L \cdot d_p(x, x')] \geq 1 - \beta \quad (11)$$

Note that equation 11 is true for each feature $i \in \{1, ..., d\}$. To conclude our proof, we note that

$$d_p(x, y) = \sqrt[p]{\sum_i^d |x_i - y_i|^p} \leq \sqrt[p]{\sum_i^d \max_i |x_i - y_i|^p} = \sqrt[p]{d} \max_i d_p(x_i, y_i)$$

Utilizing this in equation 11, without loss of generality assuming $d_p(\phi_i(x), \phi_i(x'))$ corresponds to the maximum, gives us:

$$P[d_p(\phi(x), \phi(x')) \leq 2\sqrt[p]{d}L \cdot d_p(x, x')] \geq 1 - \beta \quad (12)$$

Since $P[d_p(\phi(x), \phi(x')) \leq 2\sqrt[p]{d}L \cdot d_p(x, x')]$ in equation 12 defines $A_{\lambda, r}$ for $\lambda = 2\sqrt[p]{d}L$, this concludes the proof. $\square$

**Corollary 1.** *If the prediction function $f$ is locally deterministically $L-$Lipschitz ($\alpha = 0$) at radius $r$ then Shapley explainers are $\lambda-$astute for radius $r \geq 0$ for $\lambda = 2\sqrt[p]{d}L$*

*Proof.* Note that definition 3 reduces to the definition of deterministic Lipschitz if $\alpha = 0$. Which means equation 12 will be true with probability 1. Which concludes the proof. $\square$

### 3.1.2 ASTUTENESS OF "REMOVE INDIVIDUAL" EXPLAINERS

Within the framework of feature removal explainers, a sub-category is the explainers that work by removing a single feature from the set of all features and calculating feature attributions based on change in prediction that result from removing that feature. This category includes Occlusion, CXPlain (Schwab & Karlen, 2019), PredDiff (Zintgraf et al., 2017) Permutation tests (Strobl et al., 2008), and feature ablation explainers (Lei et al., 2018).

"Remove individual" explainers determine feature explanations for the $i^{th}$ feature by calculating the difference in prediction with and without that feature included for a given point $x$. Let $z_{-i} \in \{0, 1\}^d$ represent a binary vector with $z_i = 0$, then the explanation for feature $i$ can be written as:

$$\phi(x_i) = f(x) - f(x \odot z_{-i}) \quad (13)$$

**Theorem 3.2.** *(Astuteness of Remove individual explainers) Consider a given $r \geq 0$ and $0 \leq \alpha \leq 1$ and a trained predictive function $f$ that is locally probabilistic Lipschitz with a constant $L$, radius $r$ measured using $d_p(.,.)$ and probability at least $1 - \alpha$. Then for Remove individual explainers, we have the astuteness $A_{r, \lambda} \geq 1 - \alpha$, for $\lambda = 2\sqrt[p]{d}L$, where $d$ is the dimensionality of the data.*

*Proof.* (Sketch, full proof in Appendix A) By considering another point $x'$ such that $d_p(x, x') \leq r$ and equation 13 we get,

$$d_p(\phi(x_i), \phi(x'_i)) = d_p(f(x) - f(x \odot z_{-i}), f(x') - f(x' \odot z_{-i})) \quad (14)$$

then following the exact same steps as the proof for Theorem 1 i.e. writing the right hand side in terms of $p$-norm, utilizing triangular inequality, and the definition of probabilistic Lipschitzness leads us to the desired result. $\square$

**Corollary 2.** *If the prediction function $f$ is locally $L-$Lipschitz at radius $r \geq 0$, then remove individual explanations are $\lambda-$astute for radius $r$ and $\lambda = 2\sqrt[p]{d}L$.*

*Proof.* Same as proof for Corollary 2.1. $\qquad\square$

### 3.1.3 ASTUTENESS OF RISE

RISE determines feature explanation for the $i^{th}$ feature by sampling subsets of features and then calculating the mean value of the prediction function when feature $i$ is included in the subset. RISE feature attribution for a given point $x$ and feature $i$ for a prediction function $f$ can be written as:

$$\phi_i(x) = \mathbb{E}_{p(z|z_i=1)}[f(x \odot z)] \tag{15}$$

The following theorem establishes the bound on $\lambda$ for *explainer astuteness* of RISE in relation to the Lipschitzness of black-box prediction function.

**Theorem 3.3.** *(Astuteness of RISE) Consider a given $r \geq 0$ and $0 \leq \alpha \leq 1$, and a trained predictive function $f$ that is locally deterministically Lipschitz with a constant $L$ ($\alpha = 0$), radius $r$ measured using $d_p(.,.)$ and probability at least $1 - \alpha$. Then for RISE explainer is $\lambda-$astute for radius $r$ and $\lambda = \sqrt[p]{d}L$.*

*Proof.* (Sketch, full proof in Appendix A)

Given input $x$ and another input $x'$ s.t. $d(x,x') \leq r$, using equation 15 we can write

$$
\begin{aligned}
d_p(\phi_i(x), \phi_i(x')) &= d_p(\mathbb{E}_{p(z|z_i=1)}[f(x \odot z)], \mathbb{E}_{p(z|z_i=1)}[f(x' \odot z)]) \\
&= ||\mathbb{E}_{p(z|z_i=1)}[f(x \odot z)] - \mathbb{E}_{p(z|z_i=1)}[f(x' \odot z)]||_p = ||\mathbb{E}_{p(z|z_i=1)}[f(x \odot z) - f(x' \odot z)]||_p
\end{aligned}
\tag{16}
$$

Using Jensen's inequality on R.H.S followed by the fact that $E[f] \leq \max f$

$$d_p(\phi_i(x), \phi_i(x')) \leq \max_z d_p(f(x \odot z), f(x' \odot z)) \tag{17}$$

Using the fact that $f$ is is deterministically Lipschitz and $d_p(\phi(x), \phi(x')) \leq \sqrt[p]{d} * \max_i d_p(\phi_i(x), \phi_i(x'))$ gives us,

$$P[d_p(\phi(x), \phi(x') \leq \sqrt[p]{d}L \cdot d_p(x, x')] \geq 1 \tag{18}$$

Since $P[d_p(\phi(x), \phi(x') \leq \sqrt[p]{d}L \cdot d_p(x, x')]$ defines $A_{\lambda,r}$ for $\lambda = \sqrt[p]{d}L$, this concludes the proof. $\quad\square$

### 3.2 IMPLICATIONS

The above theoretical results all provide the same critical implication, that is, explainer astuteness is lower bounded by the Lipschitzness of the prediction function. This means that black-box classifiers that are locally smooth (have a small $L$ at a given radius $r$) lend themselves to probabilistically more robust explanations. This work provides the theoretical support on the importance of enforcing smoothness of classifiers to astuteness of explanations. Note that while this implication makes intuitive sense, proving it for specific explainers is non-trivial as demonstrated by the three theorems above. The statement holds true for all three explainers when the classifier can be assumed to be deterministically Lipschitz, the conditions under which it is still true for probabilistic Lipschitzness vary in each case. For Theorem 3.1 we have to assume that distribution $\mathcal{D}$ is defined over masked data in addition to the input data and ideally the probability of sampling of masked data from is significantly smaller compared to probability of sampling points with no value exactly equal to 0. For Theorem 3.2 the statement is true without additional assumptions. For Theorem 3.3 we can only prove the statement to be true for the detereminsitic case.

## 4 EXPERIMENTS

To demonstrate the validity of our theoretical results, we perform a series of experiments. We train four different classifiers on each of five datasets, and then explain the decisions of these classifiers using three explainers.

We utilize three simulated datasets introduced by (Chen et al., 2018) namely *Orange Skin*(OS), *Nonlinear Additive*(NA) and *Switch*, and two real world datasets from UCI Machine Learning

repository (Asuncion & Newman, 2007) namely *Rice* (Cinar & Koklu, 2019) and *Telescope* (Ferenc et al., 2005). Details for these datasets can be found in Appendix B.

For each dataset we train the following four classifiers; **2layer**: A two-layer MLP with ReLU activations. For simulated datasets each layer has 200 neurons, while for the 2 real datasets we use 32 neurons in each layer. **4layer**: A four-layer MLP with ReLU activations, with the same number of neurons per layer as *2layer*. **linear**: A linear classifier. **svm**: A support vector machine with Gaussian kernel. The idea here is that each of these classifiers will have different Lipschitz behavior, and that should lower bound the explainer astuteness when explaining each of these classifiers according to our theoretical results.

We evaluate 3 explainers here that are representative of our 3 theorems. Gradient based approximation and kernel shap approximation of **SHAP**(Lundberg & Lee, 2017) for the NN classifiers and SVM respectively serve as representative of Theorem 3.1. Both are included in the implementation provided by the authors. We modify the implementation of **RISE**(Petsiuk et al., 2018) provided by the authors for image datasets to work with tabular datasets, this serves as representative for Theorem 3.3. Implementation provided by authors for **CXPlain** (Schwab & Karlen, 2019) serves as representative for Theorem 3.2.[1].

## 4.1 EFFECT OF LIPSCHITZ CONSTRAINTS ON EXPLAINER ASTUTENESS

Following Gouk et al. (2021)'s proposal we constrain the Lipschitz constant for each layer by adding a projection step during training where after each update the weight matrices are projected to a feasible set if they violate the constraints on the Lipschitz constant, the constraints can be controlled via a hyperparameter. We use this method to train a four layer MLP with high, low and no Lipschitz constraint. We then calculate astuteness of each of our explainers for all three versions of this neural network. Figure 2 shows the results. The goal of this set of experiments is to demonstrate the relationship between Lipschitz regularity of a NN and the astuteness of explainers. As the *same* NN is trained on the *same* data but with different levels of Lipschitz constraints enforced, the astuteness of explainers varies accordingly. In all cases we see astuteness reaching 1 for smaller values of $\lambda$ for the same NN when it is highly constrained (lower lipschitz constant $L$) vs less constrained or unconstrained. *The results provide empirical evidence in support of the main conclusion that can be drawn from our work: i.e., enforcing Lipschitzness on classifiers lends them to more astute post-hoc explanations.*

## 4.2 ESTIMATING PROBABILISTIC LIPSCHITZNESS AND LOWER BOUND FOR ASTUTENESS

To demonstrate the connection between explainer astuteness and probabilistic Lipschitzness as alluded to by our theory we need to estimate probabilistic Lipschitzness for classifiers. In our experiments we achieve this by by empirically estimating the $\mathbb{P}_{x,x'\sim\mathcal{D}}$ (equation 3) for a range of values of $L \in (0,1)$ incremented at 0.1. We do this for each classifier and for each dataset $D$ and set $r$ as median of pairwise distance for all training points. According to equation 3 this gives us an upperbound on $1 - \alpha$ i.e. we can say that for a given $L, r$ the classifier is Lipschitz with probability at least $1 - \alpha$. We can use the estimates for probabilistic Lipschitzness to predict the lower bound of astuteness using our theorems. We do this by noting that our theorems imply that for $\lambda = CL\sqrt{d}$, explainer astuteness is $\geq 1 - \alpha$. This means we can guarantee that for $\lambda \geq LC\sqrt{d}$ explainer astuteness should be lower bounded by $1 - \alpha$. For each dataset-classifier-explainer combination we can plot two curves. One, that represents the predicted lower bound on explainer astuteness given a classifier, as described in the previous paragraph. Second, the actual estimations of explainer astuteness using Definition 2. According to our theoretical results, at a given $\lambda$ the estimated explainer astuteness should stay above the predicted astuteness based on the Lipschitzness of classifiers. We show these curves in Appendix Figure 3 but summarize them in tabular form in Table 1 to conserve space. The table shows the difference between the AUC under the estimated astuteness curves (**AUC**) and the AUC under the predicted lower bound (**AUC$_{lb}$**). This number captures the average difference of the lowerbound over a range of $\lambda$ values. *Note that the values are all positive supporting our result as a lower bound.*

---

[1]SHAP: https://github.com/slundberg/shap, RISE: https://github.com/eclique/RISE, CXPLAIN: https://github.com/d909b/cxplain

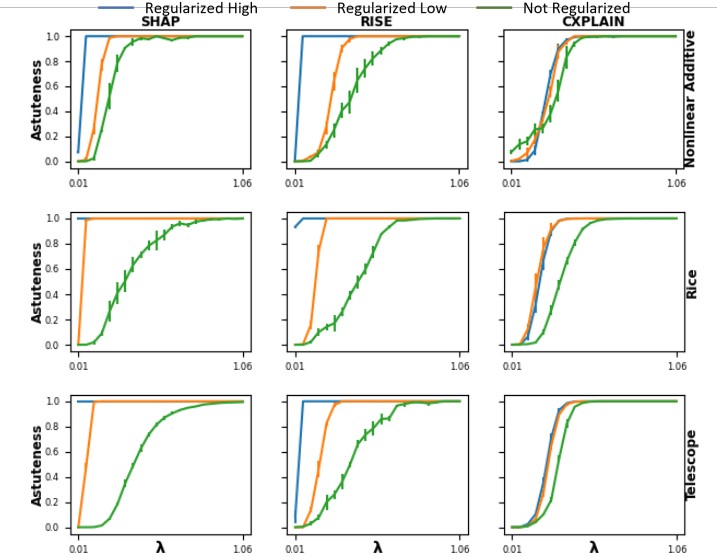

Figure 2: Regularizing the Lipschitness of a neural network during training results in higher astuteness for the same value of $\lambda$. Higher regularization results in lower Lipschitz constant (Gouk et al., 2021). Astuteness reaches 1 for smaller values of $\lambda$ with Lipschitz regularized training, as expected from our theorems. The errorbars represent results across 5 runs to account for randomness in explainer runs.

Table 1: $\mathbf{AUC} - \mathbf{AUC_{lb}}(\downarrow)$. The observed AUC is lower bounded by the predicted AUC. As expected, The difference between the two is always $\geq 0$.

| Datasets | 2layer | | | 4layer | | | linear | | | svm | | |
|---|---|---|---|---|---|---|---|---|---|---|---|---|
| | SHAP | RISE | CXPlain | SHAP | RISE | CXPlain | SHAP | RISE | CXPlain | SHAP | RISE | CXPlain |
| OS | .585 | .477 | .551 | .489 | .415 | .426 | .043 | .017 | .043 | .761 | .628 | .732 |
| NA | .359 | .289 | .318 | .285 | .216 | .244 | .452 | .391 | .474 | .742 | .653 | .708 |
| Switch | .053 | .053 | .003 | .086 | .083 | .039 | .043 | .028 | .034 | .557 | .472 | .524 |
| Rice | .249 | .142 | .229 | .292 | .131 | .252 | .258 | .165 | .241 | .426 | .347 | .413 |
| Telescope | .324 | .213 | .317 | .345 | .244 | .333 | .223 | .149 | .211 | .501 | .439 | .504 |

## 5 CONCLUSION, LIMITATIONS AND BROADER IMPACT

In this paper we formally defined *explainer astuteness* which captures the probability that a given explainer will assign similar explanations to similar points. We theoretically prove that this explainer astuteness is proportional to the *probabilistic Lipschitzness* of the black-box function that is being explained. As probabilistic Lipschitzness captures local smoothness properties of a function, this result suggests that enforcing smoothness on black-box models can lend these models to more robust explanations. In terms of limitations, we observe that our empirical results suggest that our predicted lower bound can be tightened further. One possible conjecture here is that the tightness of this bound depends on how different explainers calculate attribution scores, e.g. empirically we observe RISE and SHAP (that both depend on expectations over subsets) behave similarly to each other but different from CXPlain. Some explainers such as LIME for tabular data have the option to use a discretization step prior to calculating feature attributions. As a consequence, two observations with all features belonging to the same bins would receive exactly the same explanation, whereas two arbitrarily close inputs may receive completely different explanations (when the number of perturbed sample is large (Garreau & von Luxburg, 2020)). In that sense, tabular LIME would not be astute by our formulation, regardless of classifier Lipschitzness. Robustness is also only one property of a reliable explainer; there are other properties that are investigated in recent literature, as we outline in Section 1. These other properties, e.g. faithfulness (Agarwal et al., 2022) may also be theoretically probed in very similar ways as we did for robustness. Additionally, robustness can sometimes be at odds with with correctness (See for example Zhou et al. (2022) and "Logic Trap 3" in Ju et al. (2022)) and is best viewed as one part of explanation reliability and trustworthiness (Zhou et al., 2022).

From a broader societal impact perspective, we would like to make it clear that just enforcing Lipschitzness on blackbox classifiers should not be considered as doing enough in terms of making them more transparent and interpretable. Our work is intended to be a call to action for the field to concentrate more on improving blackbox models for explainability purposes when they are conceptualized and trained and provides one of possibly many ways to achieve that goal.

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

## A    DETAILED PROOFS

We include the detailed proofs for Lemma 1, Theorems 3.3 and 3.2 here.

*Proof.* (Proof for Lemma 1)

Let us assume,

$$p_k = P[\mathbb{N}_k], s.t. \mathbb{N}_k = \{x \mid x \in \mathbb{R}^d, ||x||_0 = k, x_i \neq 0\}$$

and let $\hat{\mathbb{L}}$ be the set of points that violate Lipschitzness, then assume,

$$\gamma_k = P[\hat{\mathbb{L}} \mid \mathbb{N}_k]$$

given that $\alpha$ is the probability of the set of points that violate Lipschitzness across $\mathcal{D}$, we can use Bayes' rule to write,

$$\alpha = P[\hat{\mathbb{L}}] = \sum_{k=1}^d p_k \gamma_k$$

If we consider the case where the sets $\mathbb{N}_k$ are finite, each $\mathbb{N}_k$ can be mapped to a set $\mathbb{N}'_k$ of cardinality,

$$|\mathbb{N}'_k| = \sum_{b=0}^{d-k} \binom{d-k}{b} |\mathbb{N}_k| = 2^{d-k} |\mathbb{N}_k|$$

In more general terms, the probability of $\mathbb{N}'_k$ can be written as,

$$p'_k = P[\mathbb{N}'_k] = \frac{2^{d-k} p_k}{\sum_{j=1}^d 2^{d-j} p_j} = \frac{2^{-k} p_k}{\sum_{j=1}^d 2^{-j} p_j}$$

Let us define $\beta$ as the proportion of points in *all* $\mathbb{N}'_k$ that also violate Lipschitzness in their unmasked form. This leads us to the following equation for $\beta$

$$\beta = \frac{\sum_{k=1}^d 2^{-k} p_k \gamma_k}{\sum_{j=1}^d 2^{-j} p_j}$$

The worse case $\beta$ would then be obtained by considering a maximization over $\gamma_k$,

$$\beta^* = \max_{\gamma_1, \ldots, \gamma_d} \frac{\sum_{k=1}^d 2^{-k} p_k \gamma_k}{\sum_{j=1}^d 2^{-j} p_j}, \sum_{i=1}^d p_i \gamma_i = \alpha, 0 \leq \alpha \leq 1, 0 \leq \gamma_i \leq 1, \forall i = 1, \ldots, d \qquad (19)$$

This constrained optimization problem can be solved by assigning $\gamma_k = 1$ for the largest $p_k$ until the budget $\alpha$ is exhausted where only a fractional value of $\gamma$ can be assigned, and 0 for the remaining values of $k$. This $\beta^* \geq \alpha$ in general. In the specific case where $p_k \to 0$ for $k = 2, \ldots, d$, when compared to $p_1$ (i.e. where the probability of sampling a point from $\mathcal{D}$ such that any of the values are *exactly* 0 is very small compared to the probability of sampling points with all non-zero values which would generally be the case for sampling real data), $\beta^* \to \alpha$ ☐

*Proof.* (For Theorem 3.2) By considering another point $x'$ such that $d_p(x, x') \leq r$ and equation 13 we get,

$$d_p(\phi(x_i), \phi(x'_i)) = d_p(f(x) - f(x \odot z_{-i}), f(x') - f(x' \odot z_{-i})) \qquad (20)$$

using the fact that $d_p(x, y) = ||x - y||_p$ where $||.||_p$ is the $p$-norm, the RHS gives us,

$$d_p(\phi_i(x), \phi_i(x')) = ||f(x) - f(x \odot z_{-i}) - f(x') + f(x' \odot z_{-i})||_p \qquad (21)$$

using triangular inequality,

$$d_p(\phi_i(x), \phi_i(x')) \leq ||f(x) - f(x')||_p + ||f(x' \odot z_{-i}) - f(x \odot z_{-i})||_p \tag{22}$$

w.l.o.g assuming the first term on the right is bigger than the second term

$$d_p(\phi_i(x), \phi_i(x')) \leq 2||f(x) - f(x')||_p = 2d_p(f(x), f(x')) \tag{23}$$

using the fact that $f$ is probabilistic Lipschitz get us,

$$P[d_p(\phi_i(x), \phi_i(x')) \leq 2Ld_p(x, x')] \geq 1 - \alpha \tag{24}$$

to conclude the proof note that $d_p(\phi(x), \phi(x')) \leq \sqrt[p]{d} * \max_i d_p(\phi_i(x), \phi_i(x'))$, which gives us,

$$P[d_p(\phi(x), \phi(x')) \leq 2\sqrt[p]{d}L \cdot d_p(x, x')] \geq 1 - \alpha \tag{25}$$

$$\square$$

*Proof.* (For Theorem 3.3)

Given input $x$ and another input $x'$ s.t. $d(x, x') \leq r$, using equation 15 we can write

$$
\begin{aligned}
d_p(\phi_i(x), \phi_i(x')) &= d_p(\mathbb{E}_{p(z|z_i=1)}[f(x \odot z)], \mathbb{E}_p(z|z_i = 1)[f(x' \odot z)]) \\
&= ||\mathbb{E}_{p(z|z_i=1)}[f(x \odot z)] - \mathbb{E}_p(z|z_i = 1)[f(x' \odot z)]||_p \\
&= ||\mathbb{E}_{p(z|z_i=1)}[f(x \odot z) - f(x' \odot z)]||_p
\end{aligned}
\tag{26}
$$

Using Jensen's inequality on R.H.S,

$$d_p(\phi_i(x), \phi_i(x')) \leq \mathbb{E}_{p(z|z_i=1)}[||f(x \odot z) - f(x' \odot z)||_p] \tag{27}$$

Using the fact that $E[f] \leq \max f$,

$$
\begin{aligned}
d_p(\phi_i(x), \phi_i(x')) &\leq \max_z ||f(x \odot z) - f(x' \odot z)||_p \\
&= \max_z d_p(f(x \odot z), f(x' \odot z))
\end{aligned}
\tag{28}
$$

Using the fact that $f$ is deterministically Lipschitz with some constant $L \geq 0$, and $d_p(x \odot z, x' \odot z) \leq d_p(x, x'), \forall z$. Then using the definition of probabilistic Lipschitz with $\alpha = 0$ we get,

$$P(\max_z d_p(f(x \odot z), f(x' \odot z)) \leq L * d(x, x') \geq 1 \tag{29}$$

Using this in equation 28 gives us,

$$P[d_p(\phi_i(x), \phi_i(x')) \leq L * d(x, x')] \geq 1 \tag{30}$$

Note that equation 30 is true for each feature $i \in \{1, ..., d\}$. To conclude the proof note that $d_p(\phi(x), \phi(x') \leq \sqrt[p]{d} * \max_i d_p(\phi_i(x), \phi_i(x'))$. Utilizing this with equation 30 leads us to

$$P[d_p(\phi(x), \phi(x') \leq \sqrt[p]{d}L \cdot d_p(x, x')] \geq 1 \tag{31}$$

Since $P[d_p(\phi(x), \phi(x') \leq \sqrt[p]{d}L \cdot d_p(x, x')]$ defines $A_{\lambda,r}$ for $\lambda \geq \sqrt[p]{d}L$, this concludes the proof. $\square$

## B  DATASET DETAILS

- **Orange-skin**: The input data is again generated from a 10-dimensional standard Gaussian distribution. The ground truth class probabilities are proportional to $\exp\{\sum_{i=1}^{4} X_i^2 - 4\}$. In this case the first 4 features are important globally for *all* data points.

- **Nonlinear-additive**: Similar to *Orange-skin* dataset except the ground trugh class probabilities are proportional to $\exp\{-100\sin 2X_1 + 2|X_2| + X_3 + \exp\{-X_4\}\}$, and therefore each of the 4 important features for prediction are nonlinearly related to the prediction itself.

- **Switch**: This simulated dataset is specifically for instancewise feature explanations. For the input data feature $X_1$ is generated by a mixture of Gaussian distributions centered at $\pm 3$. If $X_1$ is generated from the Gaussian distribution centered at $+3$, $X_2$ to $X_5$ are used to generate the prediction probabilities according to the *Orange skin* model. Otherwise $X_6$ to $X_9$ are used to generate the prediction probabilities according to the *Nonlinear-additive* model.

- **Rice** (Cinar & Koklu, 2019):This dataset consists of 3810 samples of rice grains of two different varieties (*Cammeo* and *Osmancik*). 7 morphological features are provided for each sample.

- **Telescope**(Ferenc et al., 2005): This dataset consists of $19000+$ Monte-Carlo generated samples to simulate registration of high energy gamma particles in a ground-based atmospheric Cherenkov gamma telescope using the imaging technique. Each sample is labelled as either background or gamma signal and consists of 10 features.

## C  TRAINING DETAILS

Training splits and hyperparameter choices have relatively little effect on our experiments. Regardless, the details used in results shown are provided here for completeness:

- **Train/Test Split:** For all synthetic datasets we use $10^6$ training points and $10^3$ test points. The neural networks classifiers were trained with a batch size of 1000 for 2 epochs. While SVM was trained with default parameters used in https://scikit-learn.org/stable/modules/generated/sklearn.svm.SVC.html.

  For Telescope and Rice datasets test set sizes of $5\%$ and $33\%$ were used, with a batch size of 32 trained for 100 epochs. SVM was again trained with default parameters.

- **radius $r$:** For all experiments we used radius equal to the median of pairwise distance. This is standard practice and also allows for a big enough $r$ where we can sample enough points to provide empirical estimates.

## D  ADDITIONAL RESULTS

Table 2 shows the normalized AUC for the estimated explainer astuteness and the predicted AUC based on the predicted lower bound curve. As expected the predicted AUC lower bounds the estimated AUC.

Figure 3 shows the same plots as shown in Figure **??** but includes all datasets.

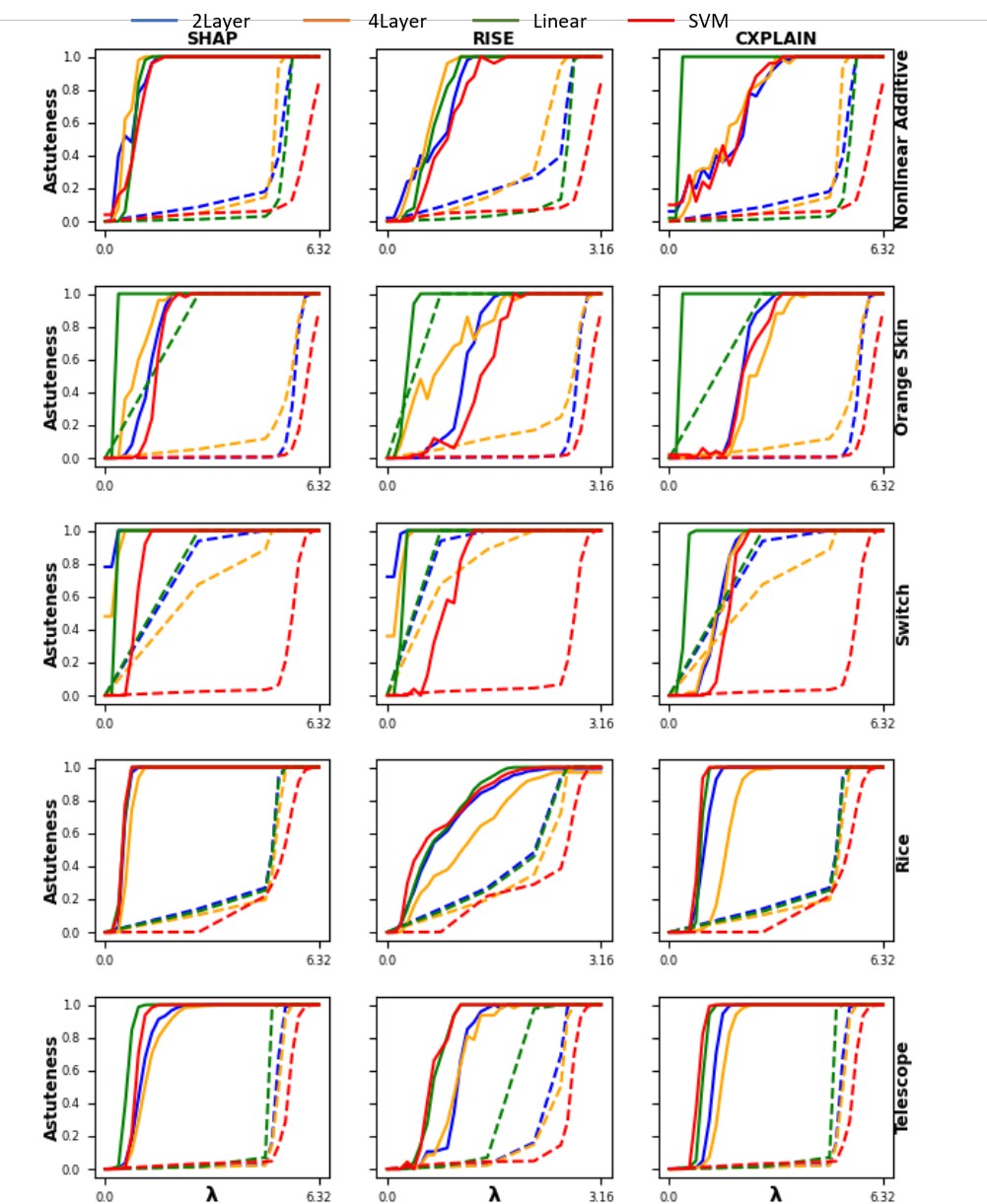

Figure 3: This figure experimentally shows the implication of our theoretical results. It corresponds to the AUC values shown in Table 1. Given each combination of dataset, classifier and explainer we observe that the estimated explainer astuteness for SHAP, RISE and CXPLAIN is lower bounded by the astuteness predicted by our theoretical results given a value of $\lambda$. The predicted lower bound is depicted by dashed lines, while solid lines depict the actual estimate of explainer astuteness.

Table 2: **Observed AUC and (Predicted AUC)**. The observed AUC is lower bounded by the predicted AUC and so the observed AUC should always be higher than the predicted AUC. The AUC values are normalized between 0 and 1.

| | 2layer | | | | 4layer | | | | linear | | | | svm | | | |
|---|---|---|---|---|---|---|---|---|---|---|---|---|---|---|---|---|
| Datasets | SHAP | RISE | CXP | (LB) | SHAP | RISE | CXP | (LB) | SHAP | RISE | CXP | (LB) | SHAP | RISE | CXP | (LB) |
| OS | .954 | .847 | .920 | (.369) | .969 | .896 | .906 | (.480) | .994 | .967 | .994 | (.950) | .945 | .813 | .917 | (.184) |
| NA | .978 | .909 | .936 | (.618) | .981 | .926 | .940 | (.696) | .972 | .912 | .994 | (.520) | .971 | .883 | .937 | (.229) |
| Switch | .998 | .996 | .948 | (.945) | .996 | .988 | .948 | (.909) | .994 | .978 | .988 | (.950) | .969 | .885 | .936 | (.412) |
| Rice | .962 | .886 | .974 | (.803) | .932 | .824 | .932 | (.793) | .968 | .901 | .962 | (.800) | .981 | .906 | .970 | (.715) |
| Telescope | .962 | .863 | .954 | (.637) | .955 | .863 | .944 | (.610) | .980 | .906 | .967 | (.756) | .969 | .909 | .972 | (.467) |

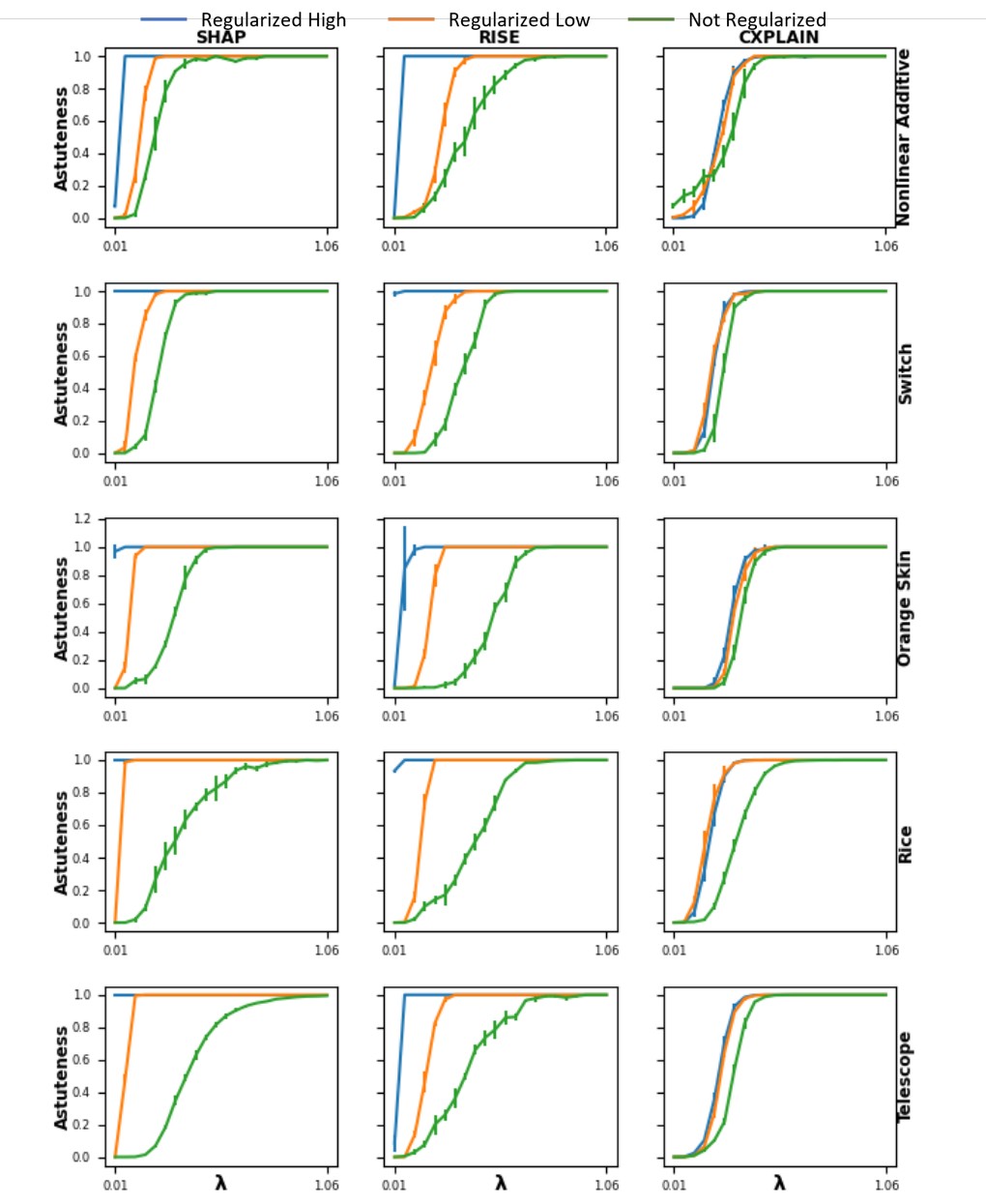

Figure 4: Regularizing the Lipschitness of a neural network during training results in higher astuteness for the same value of $\lambda$. Higher regularization results in lower Lipschitz constant (Gouk et al., 2021). Astuteness reaches 1 for smaller values of $\lambda$ with Lipschitz regularized training, as expected from our theorems. The errorbars represent results across 5 runs to account for randomness in training.

