# OpenReview forum: "Analyzing the Effects of Classifier Lipschitzness on Explainers"
_ICLR.cc/2023/Conference — Submitted to ICLR 2023_

### Official Review · Reviewer_Hif5 · 2022-10-17

**Confidence:** 4
**Correctness:** 3
**Technical Novelty And Significance:** 3
**Empirical Novelty And Significance:** 2
**Recommendation:** 5

**Clarity, Quality, Novelty And Reproducibility:**

**Clarity**

The main idea and hypothesis of the paper is clear.

**Quality**

The paper is not well written. Especially, the authors have significantly changed the ICLR latex template by removing spaces between the sections and subsections.

**Novelty**

The paper presents incremental novelty.

**Strength And Weaknesses:**

**Strengths**

1. The paper presents a theoretical analysis of the stability of explanations generated using a variety of diverse explanation methods, including SHAP, RISE, and CXPlain.
2. Theoretical and empirical analysis show that locally smooth predictive models lead to more robust explanations.

**Weaknesses and Open Questions**

1. The notion of stability is limited to the prediction behavior of the model, defined in Alvarez-Melis & Jaakkola et al. However, recently Agarwal et al. highlighted that the existing stability definition assumes model $f$ has the same behavior for similar inputs $\mathbf{x}$ and $\mathbf{x}'$, i.e., the model uses the same logic path (e.g., activating same neurons in a deep neural network) to predict the same label for the original and perturbed instance. It would be great if the authors can comment on this property of relative stability.
2. Previous works like Alvarez-Melis & Jaakkola et al., Agarwal et al., and Fel et al. have used the predictive model's Lipschitz constant to quantify the stability of explanations. This limits the novelty of the proposed work.
3. Unlike RISE and SHAP, we do not observe higher astuteness for the CXplain explanation method. It would be beneficial for the readers if the authors can give any insights into this behavior.


**References**
1. Fel, T., Vigouroux, D., Cadène, R. and Serre, T. How good is your explanation? algorithmic stability measures to assess the quality of explanations for deep neural networks. In WACV, 2022.
2. Agarwal, C., Zitnik, M. and Lakkaraju, H. Towards a rigorous theoretical analysis and evaluation of gnn explanations. In AISTATS, 2022.
3. Alvarez-Melis, D. and Jaakkola, T.S. On the robustness of interpretability methods. In arXiv, 2018.

**Summary Of The Paper:**

With the increase in the complexity of machine learning models, it becomes crucial to explain model decisions and increase transparency between human-model interactions. To this end, several explanation methods have been proposed in recent literature for explaining model predictions, but few works discuss necessary desiderata (faithfulness, stability, and fairness) to evaluate the reliability of an output explanation. In this work, the authors discuss the stability property of an explainer, i.e., an explainer should give similar explanations for similar data inputs. The paper formalizes explainer astuteness by leveraging probabilistic Lipschitzness (the probability of local smoothness of a function). Further, it provides bounds for guarantees on the astuteness of these explainers given the Lipschitzness of the prediction function and empirically validates the behavior using real-world datasets.

**Summary Of The Review:**

The paper provides interesting theoretical and empirical stability analysis of diverse explanation methods. The proof for the Astuteness of explainers that remove individual features should scale to a plethora of feature attribution methods, but the paper does not show the results for any.

---

> ### Author Response · Authors · 2022-11-15
> **Response to Reviewer Hif5**
>
> Thank you for taking the time to review our paper and raising open questions and appreciating the theoretical contributions of the paper. Please find below point wise response to your comments, we hope that these clarifications will be to your satisfaction but will be happy to follow up with further discussion.
>
> _W1: The notion of stability is limited to the prediction behavior of the model, defined in Alvarez-Melis & Jaakkola et al. However, recently Agarwal et al. highlighted that the existing stability definition assumes model  has the same behavior for similar inputs  and , i.e., the model uses the same logic path (e.g., activating same neurons in a deep neural network) to predict the same label for the original and perturbed instance. It would be great if the authors can comment on this property of relative stability._
>
> The Agarwal et al. paper discusses faithfulness (i.e. relative to logic path) as a separate desired property from stability. As we discuss in our conclusion, astuteness is but one of the many desired properties in an explainer. The goal of our paper is to theoretically connect astuteness (which is related to the “stability” property discussed by Agarwal et al.) to the probabilistic Lipschitzness of classifiers. One could conceivably explore similar theoretical connections between faithfulness of explainers to some property of classifiers as well, that however would be out of the scope of this work.
>
> _W2: Previous works like Alvarez-Melis & Jaakkola et al., Agarwal et al., and Fel et al. have used the predictive model's Lipschitz constant to quantify the stability of explanations. This limits the novelty of the proposed work._
>
> We discuss and cite Alvarez-Melis & Jaakkola et al. and Agarwal et al. as related work in the introduction section. Please see 5th paragraph in the introduction section starting with “Our work is closely related to Alverez-Melis & Jaakkola and Agarwal et al. on explainer robustness…”. We will include Fel et al. paper in the related works section as well but will like to point out that Fel et al. still only _quantifies_ some notion of explainer robustness, it does not relate explainer robustness to classifier Lipschitzness, which is the key contribution of our work.
>
> _W3: Unlike RISE and SHAP, we do not observe higher astuteness for the CXplain explanation method. It would be beneficial for the readers if the authors can give any insights into this behavior._
>
> As we discuss in the conclusions section our empirical results do suggest that the bounds we found can be tightened further. This does seem to be the case more for RISE and SHAP, than for CXplain. Our conjecture is that differences in looseness of the bound vary based on how different explainers calculate attribution scores, e.g. RISE and SHAP both rely on expectation over subsets resulting from permutations of included feature while remove explain methods like CXPlain doesn’t, that _could_ be the reason behind the difference. Nevertheless, further work is indeed required to explore the comparative looseness of the bounds.

---

### Official Review · Reviewer_tAqb · 2022-10-21

**Confidence:** 3
**Correctness:** 4
**Technical Novelty And Significance:** 3
**Empirical Novelty And Significance:** 3
**Recommendation:** 6

**Clarity, Quality, Novelty And Reproducibility:**

### Clarity, Quality, Novelty
The research motivation as well as main results are stated clearly.
Formalizing the theoretical framework connecting the smoothness of the function and the robustness of the explainers will be the novelty of the paper.

### Reproducibility
Experimental setups are described briefly in the paper.
The results may be reproducible.

**Strength And Weaknesses:**

### Strength
This will be the first study connecting the smoothness of the target function and the robustness of the explainers.
As the authors stated as "the statement is intuitive", the results look reasonable; for less smooth functions with sharp changes, the explainers also need to change drastically to reflect the change of the function, leading to less robust explanations.
Providing rigorous proofs to intuitions will be an important step towards understanding the relationship between the models and the explainers.

### Weakness
(My comment here is not really a weakness, but a suggestion for a further improvement.)
The weakness of the paper is that all the explainers considered are found to be $O(\sqrt[p]{d}L)$-Lipschitz.
Although dedicated proof techniques are required for different explainers, the results are not very surprising.
The results will be far more interesting if there are explainers that have different orders of Lipschitzness.
For example, if an explainer with large Lipschitzness is found, the use of that explainer should be avoided.
Or, if an explainer with small Lipschitzness is found, that explainer can be a recommended choice in practice.
I wonder whether there are any explainers with these interesting properties.

**Summary Of The Paper:**

This paper provides a theoretical analysis on the robustness of explainers.
Here, the robustness is measured as the difference between the two point $x$ and $x'$ in a close neighborhood.
If explanations for $x$ and $x'$ largely differ, the explainer is deemed to be not robust.
For the analysis, the authors adopted the notion of *astuteness* which was originally introduced for measuring the robustness of classifiers.
The authors extended the idea of astuteness for explainers and defined the explainer astuteness (larger the better), which is the robustness measure considered in this paper.
The authors derived the lower bounds of the explainer astuteness for some popular explainers, such as SHAP, feature removal, and RISE.
The results indicate that if the target function to be explained is $L$-Lipschitz with respect to the $\ell_p$ distance (more formally, locally probabilistic Lipschitz), the explainers are $O(\sqrt[p]{d}L)$-Lipschitz with high probability.


**Summary Of The Review:**

This is an interesting study connecting the smoothness of the function and the robustness of the explainers.
Although the results are not very surprising, establishing a theoretical framework would be an important contribution to the field.
The current paper analyzed a few popular explainers and concluded that they are all $O(\sqrt[p]{d}L)$-Lipschitz.
The results will be far more interesting if there are explainers that have different orders of Lipschitzness.

---

> ### Author Response · Authors · 2022-11-15
> **Response to Reviewer tAqb**
>
> Thank you for appreciating the usefulness of our paper and proofs and for recommending it for acceptance. We are glad that you agree that “Providing rigorous proofs to intuitions will be an important step towards understanding the relationship between the models and the explainers”, this is exactly the line of thinking we are trying to go with.
>
> We appreciate your suggestion for looking for explainers that have vastly different bounds on astuteness and to develop methods that can lead to using this for recommendation of one explainer over another and agree with it, we believe this line of thinking is extremely useful as a question of further research.

---

### Official Review · Reviewer_bjVt · 2022-10-29

**Confidence:** 4
**Correctness:** 3
**Technical Novelty And Significance:** 2
**Empirical Novelty And Significance:** 2
**Recommendation:** 3

**Clarity, Quality, Novelty And Reproducibility:**

Clarity:

The technical results are written in mathematically imprecise language, round up of this will make the contributions less messy.

The paper structure is somewhat strange. A single subsection (3.1) is larger than the rest of the paper.


Originality:

While a formal theorem relating explainer smoothness to blackbox function smoothness might be novel, it is not particularly surprising or immediately useful.



**Strength And Weaknesses:**

Strengths:

The paper has an easy-to-follow structure and repeatedly emphasizes its message relating the model Lipshitzness and explainer astuteness.

Weakness:

It is not at all clear why the end user of a blackbox explainer cares about the robustness of the explainer. The blackbox model is typically not under the end-user's control, and the only thing expected of an explainer is its faithfulness to the blackbox model. These results would be interesting, if different explainers had drastically different robustness for the same blackbox model -- that way the paper can be viewed as an advisory on what types of explainers to use in case the user cared about smoothness of the explainer.

The technical results are hard to follow and not written mathematically precisely. e.g. in Lemma 1 what are these "sampling points"? What is "i" in the set N_k? The usage of "approaching" is confusing -- what is the limit or sequence here? It looks like the set N_k is the set of points with k non-zero co-ordinates. Does the statement mean that beta=alpha when the data points all contain only one non-zero entry? Elsewhere in the paper something else is mentioned that confuses me further.

As the authors themselves mention, it is not at all surprising that the Lipshitz constant of the learned model affects the Lipschitz constant of the explainer, as indeed the explainer is designed and required to mimic the black-box model.




**Summary Of The Paper:**

The paper proposes smoothness/robustness of explainers of blackbox classifiers as a desirable objective. It gives theoretical bounds for this measure of robustness, (called explainer astuteness) in terms of the Lipschitz constant of the learned blackbox model. Theorems relating the two are given for three types of explainers, and the results are demonstrated empirically on a few datasets.

**Summary Of The Review:**

Summary:

1. The significance of explainer astuteness to the end-user needs validation.
2. The mathematical details are imprecise.
3. A method for modifying existing explainers, with increased astuteness without sacrificing faithfulness to the black-box model is required  to bump this paper to the next level
4. A result showing the theoretical maximum astuteness of any explainer for a given blackbox model (with appropriate faithfulness) would also be appreciated.

Comments:

If explainer astuteness is proposed to be of independent interest, a subjective study is required where test subjects rank different explainers based on repeated use of the blackbox model and the explainer.

---

> ### Author Response · Authors · 2022-11-15
> **Response to Reviewer bjVt**
>
> Thank you for providing feedback on our paper and pointing towards interesting areas of further research. Please find below pointwise responses to the three points you raised, we hope that these clarifications will be to your satisfaction but will be happy to follow up with further discussion.
>
> _W1: It is not at all clear why the end user of a blackbox explainer cares about the robustness of the explainer. The blackbox model is typically not under the end-user's control, and the only thing expected of an explainer is its faithfulness to the blackbox model. These results would be interesting, if different explainers had drastically different robustness for the same blackbox model -- that way the paper can be viewed as an advisory on what types of explainers to use in case the user cared about smoothness of the explainer._
>
> Why should end users care about the robustness of the explainer? Because a non-robust explainer could provide them vastly different explanations for very similar inputs, such explanations will be hard to make sense of or use. The second paragraph in the introduction section provides the motivation for this including the example of two very similar patients whose diagnosis is explained by very different factors by a robust explainer, such explanations will not be useful for the clinician and may result in the clinician arriving at the wrong conclusion about the factors behind a particular diagnosis.  See also Alvarez-Melis & Jaakkola, 2018 and Ghorbani et al. 2019 citations that provide further motivation behind explainer robustness.
>
> This paper is not expecting the end user of explainers to do anything about existing explainers, it is meant to provide motivation to those who train blackbox classifiers to put smoothness constraints on the classifiers at training time so that they can lend themselves to robust explanations afterwards.
>
> _W2: The technical results are hard to follow and not written mathematically precisely. e.g. in Lemma 1 what are these "sampling points"? What is "i" in the set N_k? The usage of "approaching" is confusing -- what is the limit or sequence here? It looks like the set N_k is the set of points with k non-zero co-ordinates. Does the statement mean that beta=alpha when the data points all contain only one non-zero entry? Elsewhere in the paper something else is mentioned that confuses me further._
>
> ‘i’ is the “i-th” feature. N_k is the set of points with ‘k’ non-zero features and i-th feature being non-zero as well.
> We agree that “approaching” here can be replaced with “is approximately”. We’d also like to point out that the detailed proof for the lemma is in the appendix.
>
> _W3: As the authors themselves mention, it is not at all surprising that the Lipshitz constant of the learned model affects the Lipschitz constant of the explainer, as indeed the explainer is designed and required to mimic the black-box model._
>
> It is unclear where in our paper we said that the results are “not at all surprising”. In Section 3.2 we did say that “... while this implication makes intuitive sense, proving it for specific explainers is non-trivial as demonstrated by the three theorems above”. Providing mathematical proofs for things that make intuitive sense but are only true given certain assumptions should not be considered a weakness in our opinion.

---

> ### Author Response · Authors · 2022-12-05
> **Gentle Reminder**
>
> Dear Reviewer bjVt,
>
> Thank you again for your valuable comments and constructive suggestions. If you could please let us know whether our responses and the revised manuscript have addressed your concerns, and let us know if any issues remain.
>
> Thanks,

---

### Decision · Program_Chairs · 2023-01-20

**Decision:**

Reject

**Justification For Why Not Higher Score:**

There are serious presentation issues.

**Justification For Why Not Lower Score:**

N/A

**Metareview: Summary, Strengths And Weaknesses:**

The paper attempts to provide explanations which are similar for similar datapoints. Borrowing on the idea of astuteness of a classifier, nearby points should have the same predictions, it develops the idea of explainer astuteness. One of the main takeaways is locally smooth
prediction functions provide locally robust explanations.
The main issue seems to be that the paper needs to be more clear on the contributions vis-a-vis existing work. Several references were made by the referees. An extensive discussion on the merits of the proposed approach versus those references would be important to appreciate the contributions.





**Summary Of Ac-Reviewer Meeting:**

Not needed